# Emerging Rhabdoviruses and Human Infection

**DOI:** 10.3390/biology12060878

**Published:** 2023-06-17

**Authors:** James G. Shepherd, Chris Davis, Daniel G. Streicker, Emma C. Thomson

**Affiliations:** 1Centre for Virus Research, MRC-University of Glasgow, Glasgow G61 1QH, UK; chris.davis@glasgow.ac.uk (C.D.); daniel.streicker@glasgow.ac.uk (D.G.S.); 2School of Biodiversity, One Health and Veterinary Medicine, University of Glasgow, Glasgow G12 8QQ, UK

**Keywords:** *Rhabdoviridae*, human infection, viral zoonoses, rabies, lyssavirus

## Abstract

**Simple Summary:**

The viral family *Rhabdoviridae* comprises over 140 different viral species, the most well-known being the *Rabies lyssavirus*, the principle cause of rabies. There are, however, a large number of other viruses within the *Rhabdoviridae* family that are less well studied but may also represent human pathogens. Modern techniques such as metagenomic next-generation sequencing have facilitated the discovery of new and previously neglected viruses. This review describes the evidence for human infection by rhabdoviruses and highlights the gaps in our knowledge of the contribution of this large viral family to human disease.

**Abstract:**

*Rhabdoviridae* is a large viral family, with members infecting a diverse range of hosts including, vertebrate species, arthropods, and plants. The predominant human pathogen within the family is *Rabies lyssavirus*, the main cause of human rabies. While rabies is itself a neglected disease, there are other, less well studied, rhabdoviruses known to cause human infection. The increasing application of next-generation sequencing technology to clinical samples has led to the detection of several novel or rarely detected rhabdoviruses associated with febrile illness. Many of these viruses have been detected in low- and middle-income countries where the extent of human infection and the burden of disease remain largely unquantified. This review describes the rhabdoviruses other than *Rabies lyssavirus* that have been associated with human infection. The discovery of the Bas Congo virus and Ekpoma virus is discussed, as is the re-emergence of species such as Le Dantec virus, which has recently been detected in Africa 40 years after its initial isolation. Chandipura virus and the lyssaviruses that are known to cause human rabies are also described. Given their association with human disease, the viruses described in this review should be prioritised for further study.

## 1. Introduction

The family *Rhabdoviridae* constitutes a group of viruses infecting a diverse range of hosts, including vertebrates (including mammals, birds, reptiles, and fish), invertebrates, plants, fungi, and protozoans. *Rhabdoviridae* sits within the order *Mononegavirales*, with over 230 viruses currently assigned to the family [1]. The family has recently been divided into three large subfamilies: *Alpharhabdovirinae*, *Betarhabdovirinae*, and *Gammarhabdovirinae*. In addition, there are several more divergent viral genera currently unassigned to a subfamily. Viruses in *Alpharhabdovirinae* infect both vertebrates and invertebrates; those in *Betarhabdovirinae* infect plants and are transmitted by arthropod vectors. Viruses within *Gammarhabdovirinae* infect fish. Several pathogens of medical and veterinary importance exist within *Alpharhabdovirinae*, the foremost being *Rabies lyssavirus* (RABV) [2]. The routes of transmission of rhabdoviruses to humans are variable (Figure 1). Some rhabdoviruses, such as lyssaviruses, spread via direct mammal-to-mammal transmission, whilst others are transmitted by arthropod vectors or vertically. As well as classical pathogens such as RABV, there is growing evidence for other, less studied, rhabdoviruses causing human infection.

## 2. Rhabdovirus Structure, Genomic Organisation, Replication, and Taxonomy

Rhabdoviruses display a characteristic bullet-shaped morphology when visualised by electron microscopy (EM). Virions range between 100 and 430 nm in length and 45 and 100 nm in diameter [3]. The nucleocapsid is covered by a lipid envelope derived from the host cell membrane and is studded with trimeric glycoprotein (G) spikes that facilitate cell entry by endocytosis. The nucleocapsid displays helical symmetry and contains a ribonucleoprotein complex (RNP) comprising genomic RNA in close association with the nucleoprotein (N). This N-RNA complex is in turn bound to an RNA-dependent RNA polymerase (L) and a phosphoprotein (P). The envelope and nucleocapsid are separated by a tightly packed layer of matrix protein (M), which interacts with the transmembrane glycoproteins and the RNP complex in the nucleocapsid (Figure 2).

The prototypic rhabdovirus vesicular stomatitis virus (VSV) and the predominant pathogen RABV have been the focus of studies on rhabdovirus structural biology [4,5]. There is a detailed understanding of the interactions of the three structural proteins (M, G, and P) within the viral particle [6,7]. In addition, the structure of L and its interactions with P have been determined for both VSV and RABV [8,9]. Structural studies of the rhabdoviral glycoprotein in its various conformational states have demonstrated significant structural homology between the G proteins of RABV and VSV despite their low sequence identity, supported by structures of the Mokola virus (MOKV) and Chandipura virus (CHPV) glycoproteins in their post-fusion conformations [10,11]. Likewise, the M proteins of VSV and the lyssavirus Lagos Bat virus are structurally similar despite minimal sequence homology [12]. Conservation of protein structural features between divergent members of the family suggests it may be possible to develop antivirals with broad activity against different rhabdovirus genera. The initial development of broadly acting therapeutics has focused on lyssaviruses. The structural interactions between the RABV glycoprotein and several neutralising antibodies have recently been characterised, enhancing the understanding of functionally important epitopes that may assist in the development of pan-lyssavirus vaccines and infection-blocking therapeutics [13,14,15].

The basic genomic organisation is conserved across the family and consists of a single strand of negative sense RNA coding for the five structural proteins: from the 3′ to 5′ direction nucleoprotein (N), phosphoprotein (P), matrix protein (M), glycoprotein (G), and the large protein (L) [16]. The genome is flanked by complimentary 3′ and 5′ trailer regions that exhibit terminal complementarity. Individual genes are separated by short non-transcribed regions that lie between conserved transcription stop and start sequences. The genome of some members of the family includes additional transcriptional units termed accessory genes. These extra open reading frames (ORFs) either overlap established structural genes or are found in the intergenic regions between them [17,18]. While the nature of most accessory genes is poorly understood, some code for non-structural glycoproteins [19].

The replication cycle is largely consigned to the cytoplasm. The viral polymerase binds to the N-RNA complex at its 3′ end and transcribes the viral genes in decreasing abundance from the 3′ to the 5′ end of the genome by variable dissociation of the polymerase at each intergenic transcription stop signal [20,21]. The resultant viral mRNAs are monocistronic and have a 5′ cap with 3′ polyadenylation. After adequate replication of viral proteins, the polymerase switches to a replicative mode and ignores the intergenic junctions, yielding full-length positive-sense antigenomes that serve as a template for the generation of genomic N-RNA complexes. Assembly occurs in the cytoplasm, and virions are released from the cell by budding.

The family *Rhabdoviridae* has undergone significant expansion in the last decade. Viral taxonomy is increasingly determined by sequence information alone, with biological characteristics not required for phylogenetic assignment [22]. This approach for novel rhabdovirus sequence discovery by metagenomic next-generation sequencing (mNGS), combined with a re-examination of several viruses isolated in the pre-molecular era, has allowed a reorganisation of the family into three subfamilies and 40 genera [1]. Several viruses within the sub-family *Alpharhabdovirinae* from the genera *Lyssavirus*, *Tibrovirus*, *Vesiculovirus*, and *Ledantevirus* are notable for their association with human disease (Figure 3). Little is known about the biology or ecological associations of the fewer than 10 non-lyssavirus rhabdoviruses detected in humans.

## 3. *Lyssavirus*

Rabies is a neglected disease and has been estimated to cause 59,000 human deaths annually, with most cases occurring in low- and middle-income countries where RABV is endemic [23]. Rabies is a zoonosis caused by viruses of the genus *Lyssavirus*. RABV is the type species of the genus and is the major cause of human disease. Human infection is predominantly caused by bites from domesticated dogs, although bats are an important reservoir of RABV in some parts of the world [24,25]. RABV infection causes acute encephalitis that is almost invariably fatal once the illness is clinically apparent. There is no effective treatment once disease is established, and the mainstay of prevention is bite avoidance, pre-exposure vaccination in high-risk groups, and post-exposure prophylaxis comprising rabies-specific immunoglobulin and vaccination [26]. RABV is thought to have originated in bats; however, the virus is readily capable of host switching and is currently maintained in transmission cycles in carnivores such as dogs, foxes, and racoons [27]. Distinct transmission cycles in wild non-human primates are speculated but remain unconfirmed [28]. In addition to RABV, several other, mostly bat-associated lyssaviruses have been implicated in human infection (Table 1).

Human bat-associated rabies cases in Europe are caused by European bat 1 lyssavirus (EBLV-1) and European bat 2 lyssavirus (EBLV-2). EBLV-1 is hosted by the European serotine bat Eptesicus serotinus, whilst EBLV-2 has been detected in two bat species: *Myotis dasycneme* and *Myotis daubentonii*. There have been two reported human cases of EBLV-1 to date, the latest in a French male who had a background of close contact with an unspecified bat species prior to his presentation with encephalitis. Notably, in this case, the diagnosis was made post-mortem by mNGS of brain tissue [29]. The previous case occurred in Russia in 1985 [30]. The two reported cases of EBLV-2 both occurred in bat handlers [31,32].

In Africa, Duvenhage virus (DUVV) and Mokola virus (MOKV) have been implicated in human rabies. MOKV is particularly noteworthy, as vaccines against RABV do not confer protection against MOKV [33]. MOKV was first isolated in 1968 in Nigeria from the *Crocidura* species and concurrently linked to two fatal human cases [34,35]. Subsequently, MOKV has been detected across southern Africa in cats, dogs, shrews, and rodents of the genus *Lophuromys* [36]. Owing to the fact that most cases have been detected in cats, it has been suggested that rodent species that regularly interact with cats may be the reservoir [37]. DUVV was initially isolated from a human case of encephalitis following an unprovoked bite from an insectivorous bat [38]. The virus is present across southern and eastern Africa based on its subsequent detection in bats in Zimbabwe and South Africa in the 1980s and two further human cases since 2000. The second human case occurred in 2006 with a 77-year-old male in South Africa who was scratched in the face by a bat that had flown into his room [39]. He did not seek medical attention and subsequently developed fatal encephalitis. In 2007, another case occurred in which a Dutch traveller was scratched on the nose by a bat while on holiday in Kenya. She did not receive post-exposure prophylaxis and developed encephalitis two weeks following the incident [40].

The Australian bat lyssavirus (ABLV) is the only endemic lyssavirus in Australia. There have been three fatal human cases since its discovery in 1996 as part of a wildlife surveillance programme for the detection of the Hendra virus. Further surveillance of Australian wildlife suggests it predominantly circulates in fruit bats of the genus *Pteropus* and the insectivorous *Saccolaimus flaviventris* [41]. There is genomic evidence of two distinct clades corresponding to the genus of the bat host [42]. The first human case was with a wildlife handler in Queensland in 1996 [43]. She had sustained several scratches to her arm while caring for sick bats and had reportedly been bitten by an insectivorous fruit *bat, Saccolaimus flaviventris* [44]. She subsequently developed a clinical syndrome consistent with rabies and died. The second case presented with fatal encephalitis after an unusually long incubation period of 27 months following a bite from a flying fox (*Pteropus* species) in Queensland in 1998 [44]. The latest case involved an 8-year-old child who sustained a scratch from a bat and subsequently developed fatal encephalitis [45]. There is now increased population awareness of the presence of ABLV in Australian wildlife, and hundreds of Australians receive prophylaxis following bat exposures annually [46].

Irkut virus (IRKV) is a bat-associated lyssavirus that has been isolated in China and Russia from *Murina leucogaster* bats. One human case has been recorded of a 20-year-old female who was bitten on the lip by a bat in the Russian Far East [47]. IRKV may be implicated in several cases of bat-associated rabies in China [48,49]. It has been demonstrated that both full vaccination and post-exposure prophylactic regimens protect hamsters from the lethal challenge of IRKV, although more study is required to understand if this is sufficient to protect humans from IKRV infections [49].

## 4. *Ledantevirus*

Le Dantec virus (LEDV) is the type species of the genus *Ledantevirus* [1,50]. It was first isolated in 1965 from the blood of a 10-year-old girl who presented with fever and hepatosplenomegaly to the Le Dantec hospital in Dakar, Senegal. The virus was subsequently found to be a rhabdovirus based on EM features and its antigenic cross-reactivity with Keuraliba virus (KEUV), a rhabdovirus isolated from a gerbil in Senegal [51].

Sequence analysis shows LEDV to contain the five canonical rhabdovirus proteins as well as an additional open reading frame coding for an accessory gene, designated U1, between the G and L proteins, a property shared by the closely related Kern canyon virus (KCV), Vaprio virus (VAPV), and KEUV [50,52]. A further case of human infection with LEDV was suspected in a British dockyard worker in 1969. Four days following a bite from an unidentified insect while unloading a cargo ship that had travelled from Nigeria, he developed a febrile illness associated with rash and delirium. Several years later, in 1977, the same patient was referred to the Hospital for Tropical Diseases in London after developing parkinsonism. He was screened for a wide panel of potential pathogens, revealing only the presence of antibodies to LEDV in his serum [53]. The patient had never visited West Africa but had served in the army in Sudan, Egypt, and Eritrea in the 1940s. After not being detected for several decades, a case of human infection by LEDV was detected in a male in western Uganda in 2013 by metagenomic next-generation sequencing, with evidence of serological reactivity to the LEDV glycoprotein by patient serum in ELISA and pseudotype-based neutralisation assays [54]. The detection of LEDV in distant countries over several decades suggests it may have a large geographic distribution within Africa.

To date, the reservoir and vector hosts of LEDV remain unknown. A viral RNA sequence similar to LEDV was isolated from an oropharyngeal swab from *Eptesicus isabellinus*, the meridional serotine bat in Spain [55]. These bats are insectivorous, so it is unclear if this sequence represents infection of the bat itself or residual debris from an insect meal within its oropharynx. *E. isabellinus* has a range extending from southern Spain into North Africa. It is a largely sedentary species, suggesting LEDV may be established in southern Europe. KEUR, the closest relative of LEDV, was originally isolated from several rodent species in Senegal, including gerbils and *Mastomys natalensis*.

Further evidence of a link between LEDV and bats is suggested by the host associations of other viruses within *Ledantevirus*. KCV was isolated from Californian bats in the 1950s. Several more distantly related viruses within the genus were also found in bats, including Mount Elgon bat virus and Fikirini virus, both isolated in East Africa [56,57]. Some viruses within the clade have been isolated from bat ectoparasites as well as bats themselves, including Kolente virus, isolated from both bats and ticks [58]. The Kanyawara virus sequence was recently isolated from bat flies dwelling on bats in Uganda; however, it was not detected in the blood of the host bats [59]. The authors did not go on to look for serological evidence of infection, but the close association of the virus with bats is consistent with other ledanteviruses.

There is evidence of human infection by two other viruses within *Ledantevirus*. Kumasi virus (KURV) was isolated from the spleen of a fruit bat, *Eidolon helvum*, captured in an urban area of Ghana [60]. PCR screening of organs from 487 bats from the same colony revealed the presence of the KURV virus in a further 25 individuals. Serological screening of the local human population suggested 3.7% had been exposed to KURV, with workers in the zoological gardens where the colony has its main roosting site demonstrating significantly increased antibody prevalence [60]. The antigenic relationships between KURV and other members of the family have not yet been determined, so the seroprevalence results may represent cross-reactivity with other rhabdoviruses. Finally, the Nkolbisson virus was isolated from both *Eretmapodites leucopous* mosquitoes and from a human case in Cameroon in 1965 [61]. Unfortunately, there is little further information on the circumstances surrounding its identification, and it has not been detected in other vertebrate species apart from humans.

## 5. *Tibrovirus*

The most concerning episode of human disease associated with a rhabdovirus outside the genus *Lyssavirus* was related to a cluster of three cases of hemorrhagic fever in a forest region of the Democratic Republic of the Congo (DRC) in 2009 [62]. The initial two cases presented with malaise and fever before the onset of hemorrhagic symptoms and death. The third, non-fatal, case involved a healthcare worker directly involved in caring for the first two patients before also developing a hemorrhagic fever syndrome, suggesting human-to-human transmission. After testing negative for known agents of hemorrhagic fever, acute serum from the third case was subjected to mNGS, with de-novo assembly yielding the sequence of a novel rhabdovirus within the genus *Tibrovirus*. The virus was named Bas-Congo virus (BASV) after the province where it was isolated. Neutralising antibodies to the BASV glycoprotein were demonstrated in both the surviving patient and a close contact who had delivered nursing care during his acute illness. Further serological testing of the local population and PCR screening of stored sera from other outbreaks of undiagnosed hemorrhagic-fever illness in the DRC were negative. The virus failed to grow when acute serum was inoculated into various cell lines, an issue the authors attributed to inadequate cold chain infrastructure in the DRC [62].

BASV is most closely related to the Tibrogargan virus (TIBV), isolated from cattle and mosquitoes in Australia [63,64], and the Ekpoma viruses (EKV-1 and EKV-2) that were detected by mNGS in Nigeria [65]. All share a similar genome organisation, with three additional ORFs in addition to the five canonical proteins: one in the intergenic region between the G and L genes and two separating the M and G genes. The EKV-1 and EKV-2 viruses are also capable of human infection, although their pathogenic potential appears limited. Both viruses were detected in the blood of healthy human control subjects in a study using mNGS to investigate patients with an unidentified acute febrile illness [65]. A subsequent serosurvey of the local population demonstrated seropositivity rates of 5% and 45% for EKV-1 and 2, respectively, suggesting widespread human exposure [65]. Subsequently, EKV-2 was detected in a Chinese worker who had been working in Angola in 2016 and developed fevers and chills, prompting medical evacuation to China. After testing negative for a broad range of pathogens, EKV-2 was identified by mNGS [66].

## 6. *Vesiculovirus*

Chandipura virus (CHPV) was first isolated in 1965 from two patients with a mild illness characterised by fever and myalgia in the Indian city of Nagpur [67]. It sits within the genus *Vesiculovirus* alongside the well-characterised VSV and demonstrates the canonical genome arrangement conserved across the *Rhabdoviridae* [68]. Transovarial transmission of CHPV has been demonstrated experimentally in both *Phlebotomus papatasi* sand flies and *Aedes aegypti* [67,69,70,71], and it has been isolated from wild sandflies in India and Africa [72,73].

CHPV received little attention until 2003, when it was associated with several outbreaks of childhood encephalitis in India [74,75]. In the second outbreak, 12 of the 26 children had detectable CHPV RNA in acute serum samples. While CHPV was clearly present in some of these cases, the role of the virus as an agent of encephalitis has been debated [76]. Despite being labelled as encephalitis, the clinical presentation was more consistent with febrile encephalopathy or vasculitis-associated cerebral infarction. Cerebrospinal fluid analysis was unremarkable in all cases, and the available cerebral imaging suggested brain oedema owing to middle cerebral artery infarction or vasospasm rather than encephalitis. No post-mortem pathological findings have been published. If CHPV is implicated in these outbreaks, it may be that intracerebral pathology is mediated via vasculitis or febrile seizures rather than encephalitis. Survivors recovered promptly, with no lasting neurological sequelae. Additionally, when all cases of childhood stroke surrounding the first epidemic were included in a further analysis, it was found that CHPV-negative cases were more likely to develop neurological manifestations [77]. It is possible that CHPV infection in these children is only one part of a clinical phenotype influenced by other factors such as climatic conditions and malnutrition. The nutritional status of the cases was described as poor to average, and the outbreaks coincided with periods of extreme high temperatures (36–49 °C). Neutralising antibody prevalence in the healthy population was noted to be 44% in children under five, rising to 76% in those over five, and 97% in adults [75]. As with most pathogens, the interaction of the virus with variable host and environmental factors may determine the extent of pathology experienced by an individual, as demonstrated in one of the earliest reported cases of CHPV, where a child died of Reye syndrome triggered by the virus [78]. Given the high seroprevalence, it is reasonable to assume that most cases in this region are mild.

## 7. Conclusions

There is growing evidence that rhabdoviruses, in addition to those from the genus *Lyssavirus*, are capable of infecting humans (Table 2). While there have been reports of severe manifestations, as in the case of BASV, a causal link is often difficult to establish with single isolates, and further work is required to determine whether these viruses represent true pathogens or are incidental to alternative aetiologies. Even if viruses, such as CHPV and BASV, can cause severe pathology, there appears to be a wide variation in clinical phenotype, with patients demonstrating positive serology without evidence of having developed significant disease. This is a common phenomenon, even with agents considered to be highly virulent, such as the Crimean-Congo hemorrhagic fever virus or West Nile virus, where, despite the potential for lethal outcomes, seroprevalence studies indicate a significant proportion of people become infected without developing clinically recognised disease [79,80]. Even if the pathological potential is low, rhabdoviruses may be contributing to the overall burden of disease by causing mild febrile illness, particularly agents such as EKV-2 and CHPV, which appear common in some areas. If this is the case, such viruses should be considered for inclusion in local diagnostic panels, particularly given the drive to limit inappropriate antimicrobial use in the face of increasing antimicrobial resistance [81].

Whilst some rhabdoviruses appear to commonly cause human infection, others, such as LEDV, are considered to only emerge in rare spillover events from a presumed natural reservoir. Whilst there is strong evidence that vesiculoviruses such as CHPV are arthropod-vectored, the ecology of the ledanteviruses is more obscure. The detection of various ledanteviruses in humans, bats, rodents, and arthropods may indicate a broad host range within the genus. It is also possible that bat species may be the natural reservoir, with humans and other animals experiencing infection in spillover events vectored by arthropods such as bat flies or ticks. Transmission may also occur via human exposure to bat excreta, especially where bats reside in highly urbanised areas, as in the case of KURV. If this is the case, more human cases may be predicted as the population expands into new ecosystems. Further studies are required to fully understand the ecological relationships among this diverse group of viruses.

## Figures and Tables

**Figure 1 biology-12-00878-f001:**
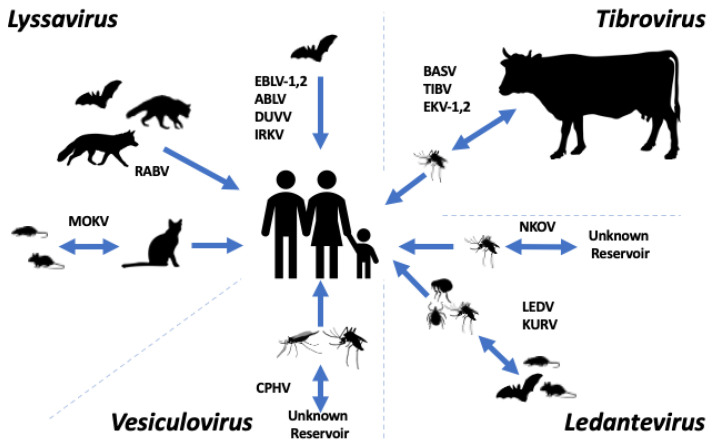
Transmission routes of selected members of the subfamily *Alpharhabdovirinae*. Depiction of the known and possible transmission routes for the members of the *Lyssavirus* genus (Rabies virus (RABV), European bat lyssavirus-1 (EBLV-1), European bat lyssavirus-2 (EBLV-2), Australian bat lyssavirus (ABLV), Duvenhage virus (DUVV), Irkut virus (IRKV), and Mokola virus (MOKV)), the *Tibrovirus* genus (Bas-Congo virus (BASV), Tibrogargan virus (TIBV), Ekpoma virus-1 (EKV-1) and Ekpoma virus-2 (EKV-2)), *Ledantevirus* genus (Le Dantec virus (LEDV), Nkolbisson virus (NKOV), and Kumasi virus (KURV), and the *Vesiculovirus* genus (Chandipura vesiculovirus (CHPV)).

**Figure 2 biology-12-00878-f002:**
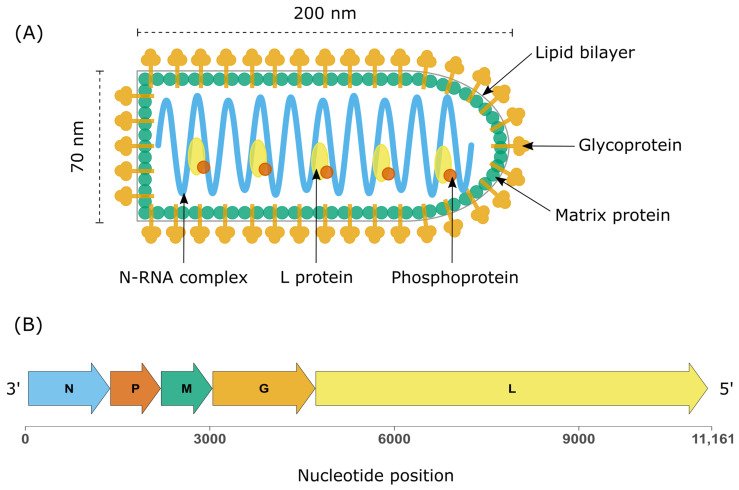
Rhabdovirus structure and genome organisation. (**A**) Structure of the prototype rhabdovirus Vesicular Stomatitis Virus (VSV). (**B**) Genome organisation of the canonical rhabdovirus VSV (accession number J02428) demonstrating the canonical genes in 3′-5′ direction: nucleoprotein (N), phosphoprotein (P), matrix protein (M), glycoprotein (G), and the large protein comprising the RDRP (L).

**Figure 3 biology-12-00878-f003:**
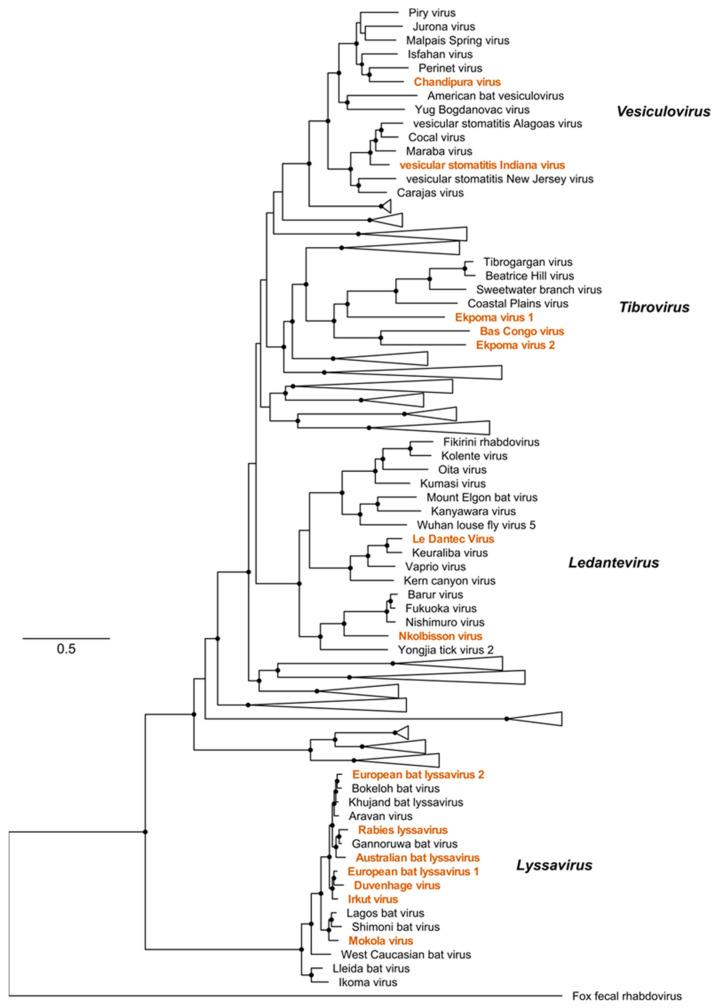
The sub-family *Alpharhabdovirinae*. Maximum likelihood phylogeny of the sub-family Alpharhabdovirinae based on a trimmed alignment of full-length L protein amino acid sequences. Viruses associated with human infection are indicated in orange text. The scale bar represents amino acid substitutions per site. Black circles indicate nodes with support values greater than 70 based on 1000 bootstrap replicates.

**Table 1 biology-12-00878-t001:** Lyssaviruses causing documented human infection.

Virus	Reservoir	Distribution
Australian bat lyssavirus	Bats	Australia
Duvenhage virus	Bats	South Africa, Zimbabwe, Kenya
European bat lyssavirus 1	Bats	Europe, Russia
European bat lyssavirus 2	Bats	Europe
Irkut virus	Bats	Russia, China
Mokola virus	Rodents	Nigeria, South Africa
Rabies virus	Various mammals	Global

**Table 2 biology-12-00878-t002:** Rhabdoviruses causing documented human infection.

Virus	Clinical Syndrome	Reservoir/Vector	Distribution	References
ABLV	Encephalitis	Bats	Australia	[41,42,43,44,45,46]
DUVV	Encephalitis	Bats	South Africa, Kenya	[38,39,40]
EBLV-1	Encephalitis	Bats	Europe, Russia	[29,30]
EBLV-2	Encephalitis	Bats	Europe	[31,32]
IRKV	Encephalitis	Bats	Russia, China	[47,48,49]
MOKV	Encephalitis	Rodents	Nigeria, South Africa	[34,35,36]
RABV	Encephalitis	Various mammals	Global	[23,24,26]
LEDV	Acute febrile illness	Unknown	Africa	[51,53,54]
NKOV	Acute febrile illness	Mosquitoes	Cameroon	[61]
KURV	Unknown	Bats	Ghana	[60]
BASV	Hemorrhagic fever	Unknown	Central Africa	[62]
EKV-1	Unknown/subclinical	Unknown	Nigeria	[65]
EKV-2	Febrile illness/subclinical	Unknown	Nigeria, Angola	[65,66]
CHPV	Fever, encephalopathy	Sandflies	India, Senegal	[67,70,71,72,73,74,75,76,77,78]

## Data Availability

Not applicable.

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
