# Peer review of "Emerging Rhabdoviruses and Human Infection"

_biology, 2023, doi:10.3390/biology12060878_

Round 1
Reviewer 1 Report
It would be useful to organize a summary-table including the virus, the vector, the illnes, the place and the reference for all these new disease-causing virus.
Author Response
We thank the reviewer for their helpful comments. We have added a table detailing each rhabdovirus that has been associated with human infection as suggested (table 2).
Reviewer 2 Report
Shepherd and colleagues review the rhabdoviruses with potential for emergence in the human population. This article is therefore particularly relevant in our post-covid era where many governmental or international organizations are seeking to establish a list of viruses to be characterized. Rhabdoviruses are very rarely included in this list.
This review is therefore also potentially interesting for decision-makers.
However, I regret that the molecular aspects are reduced to two short paragraphs in the introduction.
I regret that the authors do not even mention that the structures of the 5 canonical proteins are known for at least one rhabdovirus: that of N for VSV Indiana and RABV, that of domains of P for VSV Indiana, RABV and MOKV, that of G in different conformational states for VSV indiana, CHPV, RABV and MOKV, that of M for VSV indiana, VSV New Jersey and lagos bat virus and finally that of L for VSV Indiana and RABV.
The structures of G show taht the glycoproteins of rhabdovirus resemble each other (which suggests the possibility of rapidly making an effective vaccine in the event of emergence). The other structures provide the rational basis for designing molecules with antiviral action (potentially against distinct genera of rhabdovirus).
This should be the subject of a dedicated paragraph that would nicely complement the review.
Author Response
We thank the reviewer for their helpful comments. As suggested we have added a paragraph to the introductory section of the article (lines 67-83) outlining aspects of rhabdovirus structural biology and its relevance to therapeutic development.
Reviewer 3 Report
In the review paper entitled Emerging rhabdoviruses and human infection the authors presented the rhabdoviruses signifficant in public health. The paper is clearly prepared describing the comprehensive knowledge concerning rhabdovirus agents infecting humans. It summarize the molecular properties of rhabdoviruses, the members of Alfarhabdovirinae sub-family as well as description of its clinical cases. I highly reccomend the publication of the manuscript, however, some mistakes must be corrected:
In the abstract section in line 24 rabies-related lyssaviruses should be replaced by lyssaviruses, according the latest taksonomy.
In Fig. 2 Shimoni bat lyssavirus is missed on phylogenetic tree.
In part 3. Lyssavirus, the sentence... bats are important reservoir of human rabies should be improved. They may consist reservoir of virus....
The first sentence of the Conclusions misleads the reader because it suggests that lyssaviruses are not able to infect humans, that is not true, since there are at least 15 cases of human deaths caused by bat lyssavirus infections. I suggest to express the sentence differently or to be corrected.
Author Response
We thank the reviewer for their helpful comments. As suggested we have made the following alterations to the manuscript:
- In the abstract line 24 we have replaced “rabies-related lyssaviruses” with “lyssaviruses”.
- We have updated the phylogenetic tree in figure 1 to include Shimoni bat lyssavirus.
- In part 3 Lyssavirus we have altered the phrase “bats are important reservoir of human rabies” to read “bats are an important reservoir of RABV”.
- In the first line of the conclusion we have altered the sentence “There is growing evidence that rhabdoviruses beyond the genus Lyssavirus are capable of infecting humans” to read “There is growing evidence that rhabdoviruses in addition to those from the genus Lyssavirus are capable of infecting humans”.